**Cite this article:** Ausprey IJ. 2021
Adaptations to light contribute to the
ecological niches and evolution of
the terrestrial avifauna. *Proc. R. Soc. B* **288**:
20210853.

ecology

light, eye, bird, community assembly, niche,
evolution

**Author for correspondence:**
Ian J. Ausprey
e-mail: iausprey@ufl.edu

Electronic supplementary material is available
online at https://dx.doi.org/10.6084/m9.
figshare.c.5420181.

# Adaptations to light contribute to the ecological niches and evolution of the terrestrial avifauna

## Ian J. Ausprey

Department of Biology and Florida Museum of Natural History, University of Florida, Gainesville, FL 32611, USA

 IJA, 0000-0002-7127-2746

The role of light in structuring the ecological niche remains a frontier in understanding how vertebrate communities assemble and respond to global change. For birds, eyes represent the primary external anatomical structure specifically evolved to interpret light, yet eye morphology remains understudied compared to movement and dietary traits. Here, I use Stanley Ritland's unpublished measurements of transverse eye diameter from preserved specimens to explore the ecological and phylogenetic drivers of eye morphology for a third of terrestrial avian diversity ($N = 2777$ species). Species with larger eyes specialized in darker understory and forested habitats, foraging manoeuvres and prey items requiring long-distance optical resolution and were more likely to occur in tropical latitudes. When compared to dietary and movement traits, eye size was a top predictor for habitat, foraging manoeuvre, diet and latitude, adding 8–28% more explanatory power. Eye size was phylogenetically conserved ($\lambda = 0.90$), with phylogeny explaining 61% of eye size variation. I suggest that light has contributed to the evolution and assembly of global vertebrate communities and that eye size provides a useful predictor to assess community response to global change.

## 1. Introduction

Light is a pervasive component of ecosystems, producing the tapestry of optical sensory environments through which organisms navigate their daily lives. Despite exhaustive research on how light structures plant communities [1], the role of light in defining the ecological niche for vertebrates remains relatively enigmatic [2–4]. This is surprising given that vertebrate biodiversity persists across vast climatic and habitat gradients that vary widely in ambient light conditions [5,6]. Identifying functional traits linking light to the ecological niche will help more precisely quantify local community assembly, regional patterns in niche packing and species turnover, and the drivers of biodiversity loss in the face of global change [7–9].

Birds rely heavily on vision to detect food and predators, and the eye is the primary external anatomical trait specifically adapted to sense light [10]. For species operating at the extremes of optimal resolution, extraordinary adaptions in retinal anatomy allow individuals to hunt nocturnally (e.g. owls), target fast-moving and distant vertebrate prey (e.g. raptors), and forage in aquatic environments [11–14]. The vast majority of terrestrial avian biodiversity, however, navigates dramatically shifting mosaics of colour and luminosity that interact with structurally complex habitats [15]. Even within relatively dark forests, irradiance can vary across orders of magnitude, depending upon location within canopy strata, the position of gaps and abundance of sun flecks [5]. Despite comprehensive knowledge of ecomorphological relationships for traits related to movement and diet [16], little research has explored the one functional dimension specialized for light: the eye.

While the avian eye possesses micro-anatomical features exquisitely adapted for interpreting variation in colour and brightness, eye size is a useful trait for full community studies, because it directly relates to focal length and image resolution and can be quantified efficiently in the field or museum [3,17–19]. Specifically, larger eyes have more retinal ganglia cells, collect more light and are thought to expand the perceptual range by improving visual acuity and sensitivity to contrast [20–22]. Evolving a larger eye imposes significant trade-offs, however. Increased metabolic investment is required to maintain anatomical structures and process neurological information [23,24]. Birds have relatively large eyes and brains compared to other vertebrate taxa, posing spatial constraints within the cranium and aerodynamic consequences for maintaining flight performance [4,20,25–29]. While the pupil and optical adnexa deter optical damage in bright conditions, large eyes are thought to be more susceptible to overexposure or 'disability glare' that can impact the detection of food or predators [30,31]. Given such evolutionary trade-offs, residual variation in eye size after correcting for body mass allometry should represent an adaptive trait related to the optical environment [32].

Variation in residual eye size has been linked to multiple ecological factors for small subsets of the terrestrial avifauna. Eye size is strongly correlated with the actual light intensity micro-environments used by free-ranging birds [33] as well as several behavioural traits related to ambient light levels, including initiation of singing at dawn, arrival at feeding stations in the early morning and prevalence of nocturnal foraging [12,34–36]. Because visual acuity tends to increase with eye size, large-eyed species appear more responsive to experimental predators [37] and more likely to engage in aerial foraging manoeuvres [33,38]. The relationship between eye size and habitat disturbance is less clear, with some evidence that species with larger eyes disappear from brightly lit agricultural landscapes and forest edges [33,39]. Avian reproductive phenology is also potentially affected by the interaction of artificial light and eye morphology [40,41]. To date studies have been restricted to single communities or comparative efforts with less than 200 species, and the lack of a large-scale comparative analysis across the avian phylogeny has hampered efforts to make generalizations about the contribution of eye morphology to global patterns in community assembly.

Here, I use a dataset of eye measurements collected from preserved specimens by Stanley Ritland (SR) [32] to explore the ecological and phylogenetic correlates of eye morphology for a third of terrestrial avian diversity ($N = 2777$ species). While Ritland provided an initial draft of ecological correlates at the family level in his dissertation, I expand upon his work by (1) examining the ecological correlates of eye size at the species level within a modern phylogenetic framework, (2) including macro-ecological variables reflecting contemporary knowledge of species range sizes, latitude and life history, (3) comparing the explanatory power of eye size with a suite of morphological traits related to movement and diet, and (4) partitioning the phylogenetic and ecological contributions to eye size variation.

Regarding ecological correlates, I first hypothesized that residual eye size is correlated with habitats that constrain the quantity of light, with the prediction that species with larger eyes specialize in forests and forage in the understory where increased visual acuity is required to overcome dark foraging environments. I further predicted that increased eye size would be associated with ranges centred in the tropics, a region housing some of the darkest forests on Earth [42]. Second, I hypothesized that eye size varies by foraging behaviour and diet due to the role light plays in mediating food identification and capture, with the prediction that species employing long-distance (hyperopic) foraging manoeuvres and pursing arthropod or vertebrate prey would have larger eyes associated with heightened visual acuity. Third, I predicted that species with larger eyes would exhibit non-migratory tendencies and smaller range sizes due to less varied optical environments and the aerodynamic advantages of minimizing eye mass during long-distance flight.

## 2. Databases

### (a) Eye morphology
I extracted measurements of transverse and axial diameter (AD) from SR's unpublished dissertation [32]. All measurements were collected by SR from specimens of whole eyes preserved in formaldehyde and/or alcohol using 0.05 mm Vernier callipers. I focus on transverse diameter (TD) because (1) SR noted that his measurements of TD were more reliable than those for AD and focused on them for his dissertation, (2) TD and AD were highly correlated ($r = 0.99$; electronic supplementary material, figure S1), and (3) my analysis did not include birds of prey for which AD is considered a proxy for visual acuity at high speeds [14]. To ensure that there was no systematic difference between AD and TD, I repeated all analyses for both metrics and report the AD results in the electronic supplementary material. SR included the pre-preservation mass for most specimens measured, and I supplemented missing species with values from the Elton Traits database [43]. Sample sizes were less than or equal to five for 99% of species.

### (b) Species
I included only extant, terrestrial, diurnal and non-raptorial species. This excluded seabirds, shorebirds, wading birds, birds of prey and nocturnal/crepuscular species whose eye morphologies are adapted to aquatic foraging, extreme long-distance prey resolution and nocturnal vision, respectively [12–14]. This resulted in 2777 species from 139 families (electronic supplementary material, table S1).

### (c) Habitat
I defined habitat specialization using classifications published by Bird Life International [44]. For each species I extracted all habitats classified as 'Major', which is defined by BLI as 'required for survival'. For the 541 species lacking a 'Major' classification, I extracted habitats from the next hierarchical level of 'Suitable'. If a species was classified as having only one 'Major' habitat I considered it specialized to that specific habitat. Species with more than one 'Major' habitat were classified as habitat generalists.

### (d) Foraging behaviour
I used a recent ecomorphological analysis of the avian tree of life to define the predominate foraging manoeuvre employed by each species [16]. Specifically, I used the 'Foraging Niche' classification to categorize species as using predominately

myopic (near-sighted) versus hyperopic (far-sighted) manoeuvres (electronic supplementary material, table S2). Because the study was interested in classifying species into specialized manoeuvre types based on a combination of dietary guild and fine-scale foraging behaviour, it did not provide classifications for omnivores or species that exhibited a mix of manoeuvre strategies that did not easily conform to their fine-scale classification system. Because I was interested in a more generalized binary classification unrelated to diet or behavioural specialization, I used Birds of the World to classify those remaining species as either myopic or hyperopic ($N = 704$ species) [45]. I classified myopic manoeuvres as those that target food in the immediate visual plane (i.e. glean, pick, probe and hammer) and hyperopic manoeuvres as those that target food at a distance (i.e. sally, hawk, screen and pounce) [46]. For foraging stratum, I calculated the per cent use of the middle and upper canopy strata by combining the middle canopy, upper canopy and aerial foraging strata published in the Elton Traits database [43], creating a continuous scale ranging from 0 (ground) to 1 (canopy).

### (e) Diet
I extracted diet percentiles published in the Elton Traits database [43] and converted them to four principal component axes explaining 99% of variation in dietary preference (electronic supplementary material, table S3).

### (f) Macro-ecological variables
I extracted migratory tendency, range size and mean range latitude from a previously published global analysis of dispersal ability [47].

### (g) Morphological traits
To examine the relative importance of eye size compared to other morphological traits, I used recently published ecomorphological databases [16,47]. Specifically, I used nine principal components decomposing variation in morphospace for a suite of traits related to diet (bill length, width and depth) and movement (wing, tail, tarsus and mass), as well as the hand-wing index, an assumed correlate of dispersal ability.

### (h) Correlation of variables
All ecological variables were largely uncorrelated, and eye size was largely uncorrelated with the 10 previously published morphological axes (electronic supplementary material, table S4 and figure S2).

### (i) Phylogeny
I pruned 100 randomly selected trees downloaded from a previously published avian phylogeny [48] (Hackett backbone). Trees were based on molecular data for 2311 species (82%), and I ran all analyses across the 100 trees to account for topological uncertainty among the species with no molecular data.

## 3. Analysis
All analyses were conducted in R 4.0 using the 'phylolm', 'MASS', 'emmeans', 'visreg', 'nlme', 'ape' and 'rr2' packages [49–52].

### (a) Eye residuals
To correct for body size allometry, I extracted residuals from phylogenetic regressions of log(TD) and log(AD) on log(mass) (electronic supplementary material, figure S3A). I compared models incorporating evolutionary models for Brownian motion (BM), Pagel's $\lambda$ (PA) and an OU model of stabilizing selection, looping across 100 trees. The PA models best fit the data as ranked by AIC for TD and AD (electronic supplementary material, figure S3B), and I extracted residuals from the PA models with the median slope coefficients among the 100 trees.

### (b) Part 1: ecological correlates of eye size
I used phylogenetic multiple linear regression to examine the ecological correlates of residual eye size with all phylogenetic models repeated across 100 trees. I scaled all continuous variables to a mean of 0 and standard deviation of 1 to allow comparison among effect sizes. The following steps were completed for both TD and AD. First, I converted the following variables to binary states: habitat (forest versus non-forest), foraging manoeuvre (myopic versus hyperopic) and migratory tendency (migratory versus non-migratory). Second, I constructed a global additive model containing all variables, with interactions between each binary variable and all continuous variables. Third, I fit the global model to phylogenetic regressions for BM, PA and OU models of evolution; the PA models consistently fit best for both TD and AD (electronic supplementary material, figure S4), so I used PA models for all subsequent analyses. Fourth, I used forward and backwards AIC model selection to eliminate all variables and binary × continuous variable interactions with insufficient explanatory power. The retained variables and interactions were largely the same for TD and AD (electronic supplementary material, table S5). Fifth, I constructed phylogenetic regression models using the retained variables and binary × continuous variable interactions from the stepwise procedure to construct the final ecological regression model. Finally, I used partial residuals and the estimated marginal means and pairwise contrasts of model coefficients corrected for multiple comparisons (Tukey adjustment) to visualize and summarize the coefficients of each individual variable and interaction while controlling for all others retained in the model. Statistics exhibited little variation among trees, and I used partial residuals and statistics extracted from the trees with median coefficient estimates for subsequent inference (electronic supplementary material, figures S5 and S6). For continuous variables and interactions, inference was made on mean coefficient estimates and whether 95% confidence intervals overlapped zero.

Using residuals in subsequent linear models can create spurious results if collinearity exists among them [53]. Hence, I duplicated the initial global model using the log of absolute eye size (TD or AD) as the dependent variables and the log of mass as an explanatory variable. Results were extremely similar. I used residuals for the final analysis because they provided an intuitive index of relative trait size.

### (c) Part 2: morphological predictors of ecology
I examined the relative explanatory power of residual eye size (TD and AD) in predicting each of the previously described ecological variables compared to the 10 previously published

morphological axes related to movement and diet using phylogenetic regression. For binary dependent variables, I used phylogenetic logistic regression, which accounts for the evolution of binary traits [50]. Each model was additive and consisted of one dependent ecological variable and the 10 independent morphological traits. All traits were scaled to mean of 0 and standard deviation of 1. I ranked the predictive power of each trait by comparing the absolute value of median $z$-scores across all trees. To determine the degree to which eye size improved model fit, I calculated the coefficient of determination (pseudo $R^2$) for the 10 ecological variables used in this analysis by sequentially adding (i) phylogeny, (ii) the 10 previously published morphological trait axes and (iii) residual eye size. $R^2$ values were extracted using the R function 'R2.lik' [52]. The $R^2$ values did not vary substantially among trees, and I used the median values for inference (electronic supplementary material, figure S7).

### (d) Part 3: phylogenetic correlates of eye size and variation partitioning

I partitioned variation among the phylogenetic and ecological correlates of eye size (TD and AD) using phylogenetic and non-phylogenetic regression. For phylogenetic models, I first calculated the cumulative $R^2$ for a model incorporating a phylogenetic correlation matrix based on PA and containing no variables versus a non-phylogenetic null model. I then sequentially added variables and interactions retained from the stepwise model selection results in the following order: habitat, foraging and diet (foraging manoeuvre, stratum and dietary axes), and macro-ecology (latitude, range size and migratory tendency). Values for $R^2$ did not vary substantially among trees, and I used the median values for inference (electronic supplementary material, figure S8). To better understand how each of the 10 ecological variables contributed towards the role of phylogeny in explaining residual eye size variation, I calculated PA for each variable independently using the functions 'phylolm' (package 'phylolm') for continuous traits and 'fitDiscrete' (package 'geiger') for the three binary traits.

## 4. Results

### (a) Part 1: ecological correlates of eye size

As predicted species specializing in forests and using hyperopic foraging manoeuvres had significantly larger eyes (forest versus non-forest: $\beta = 0.036$, $p < 0.001$; hyperopic versus myopic: $\beta = 0.099$, $p < 0.001$) (figure 1; electronic supplementary material, table S6). Eye size decreased with increasing foraging strata for resident forest-dwelling species using myopic foraging manoeuvres. Migratory non-forest species with larger ranges had smaller eyes, and eye size increased with proximity to the tropics (figure 2). Results for AD were similar, except that AD did not vary with foraging stratum for any category (electronic supplementary material, figures S12 and S13).

Correlations with diet were partially mediated by habitat, foraging behaviour and migratory tendency. As predicted invertivores had larger eyes compared to herbivores. The pattern was significant only for myopic foragers based on TD and for both myopic and hyperopic foragers based on AD (figure 3; electronic supplementary material, figure S14).

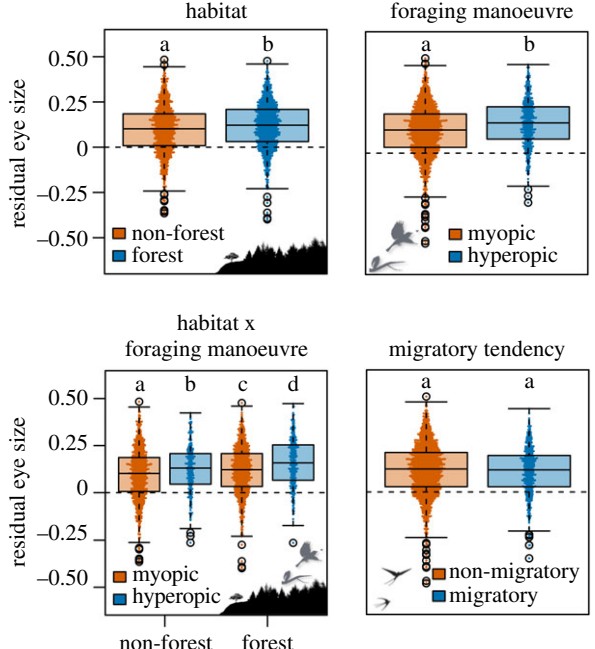

**Figure 1.** Partial residual plots for the relationship between residual eye size and habitat, foraging manoeuvre and migratory tendency for 2777 species of terrestrial birds. Boxplots represent the median, 25% and 75% quartiles, interquartile range and outliers. Pairwise contrasts corrected for multiple comparisons are significant where $p \leq 0.05$; significantly different categories are denoted by unique lower case letters.

Eye sizes of myopic invertivores were larger for forest specialists compared to non-forest species, suggesting the existence of visual constraints within darker environments. Large eyes were also associated with frugivory compared to granivory for myopic species. Finally, small and large eyes were associated with increasing amounts of nectivory and carnivory, respectively. The results for AD were similar (electronic supplementary material, figure S14).

### (b) Part 2: morphological predictors of ecology

Eye size had the largest $z$-score when predicting specialization in forests and the second largest $z$-score when predicting foraging manoeuvre, consumption of fruit versus seeds and latitude. It was the fourth ranked predictor for invertebrate versus plant consumption (figure 4). Eye size increased the total coefficient of determination represented by ecological factors for these variables by 8–28% (electronic supplementary material, table S7). Results were extremely similar for AD (electronic supplementary material, table S7 and figure S15).

### (c) Part 3: phylogenetic correlates of eye size and variation partitioning

Residual eye size was highly conserved across the avian phylogeny ($\lambda = 0.90$; figure 5), producing families with strikingly different eye morphologies (electronic supplementary material, figure S9). There was strong phylogenetic signal among residual eye size and ecological variables for the final ecological model (PA = 0.87), and phylogeny explained the majority of variation in eye size ($R^2 = 0.61$) (figure 6). Ecological variables collectively explained considerably less variation in eye size for the phylogenetic model ($R^2 = 0.09$)

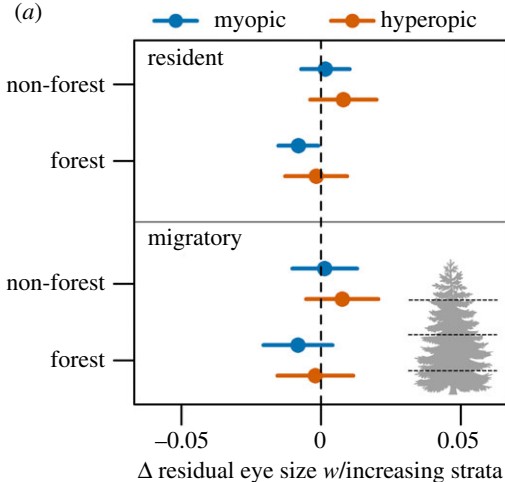

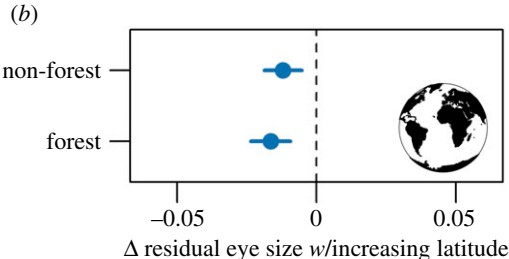

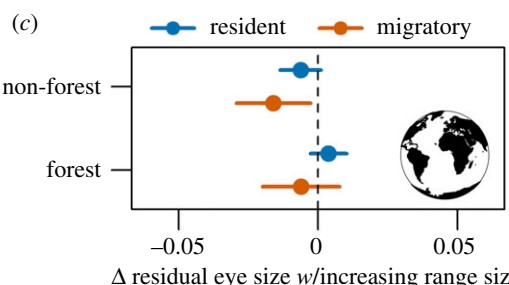

**Figure 2.** Estimated marginal means (+/− 95% CI) of model coefficients for (a) foraging stratum, (b) latitude and (c) range size in relation to residual eye size for 2777 species of terrestrial birds.

compared to the non-phylogenetic model ($R^2 = 0.41$). This was largely driven by foraging and dietary traits, which also exhibited the highest levels of phylogenetic signal among the 10 ecological variables used in this analysis (electronic supplementary material, figure S10). Results were extremely similar for AD (electronic supplementary material, figures S11 and S16).

## 5. Discussion

As predicted, eye size was strongly correlated with ecological variables related to light, such that species specializing in darker habitats and foraging manoeuvres requiring long-distance prey resolution had larger eyes. Across the globe terrestrial birds with the smallest eyes were those living in non-forested habitats and using myopic foraging manoeuvres, while species with the largest eyes specialized in forests and used hyperopic manoeuvres. Species at tropical latitudes had larger eyes, suggesting that light may contribute towards increased sensitivity to anthropogenic habitat disturbance in the tropics [54]. Despite many significant correlations with ecological variables, residual eye size was highly conserved

across the avian phylogeny, and phylogeny explained the majority of eye size variation. Strong correlations between eye size and foraging manoeuvre and diet were reduced in phylogenetic models, likely due to correlation between the evolution and ecology of those traits.

## (a) Eye morphology

The avian eye is unique among anatomical structures in being the one external trait specifically adapted to interpret light and is attuned to two interacting properties: intensity (brightness or luminosity) and wavelength (colour) [10]. Light intensity is mediated by the density and orientation of retinal cell ganglia (RCG), whereas colour is interpreted by specialized visual pigments and oil droplets found within RCG [3,13]. Although orientation of RCG and topography of the fovea contribute towards articulating precise aspects of light interpretation [18], larger eyes accommodate greater numbers of photoreceptors and improve visual acuity [55]. My results demonstrated that residual eye size correlates with multiple ecological factors related to light and can be used as a functional trait related to the optical environment.

## (b) Foraging and diet

The majority of terrestrial avian diversity encompasses a complex array of dietary and foraging strategies, all of which require mediating rapidly changing light micro-environments when identifying and capturing food [15,56,57]. Past studies of specific avian communities or small subsets of the avian phylogeny have proved inconclusive regarding the relationships between eye size and foraging behaviour [33,37,38]. Using a much larger sample of the avian phylogeny, I demonstrated that species employing myopic (near-sighted) foraging manoeuvres and species foraging in higher canopy strata that are presumably exposed to more light had smaller eyes compared to those capturing distant, mobile prey, especially at lower, darker habitat strata. The fact that hyperopic species did not vary in eye size across foraging strata suggests that the adaptive benefits associated with a longer focal length both compensate for a darker understory and outweigh potential overexposure to increased luminosity higher in the canopy. Interestingly, the association between foraging stratum and eye size was fairly weak, with a significant relationship documented only for resident forest myopic foragers and effect sizes being generally small. Birds are known to forage across widely different canopy strata [58], meaning that the highly variable light environments experienced by most species may provide weaker selective pressure on eye size than other traits, such as diet or manoeuvre.

I documented several previously unrecognized relationships between diet and eye morphology. Diet predicted eye size mainly for myopic species, which had larger eyes when consuming more fruit and invertebrate prey. However, eye size based on AD also increased with arthropod diets for hyperopic foragers, suggesting an adaptive advantage provided by increased focal length when pursuing distant prey [38]. Nectivory was strongly associated with smaller eyes, implying that the probing manoeuvres associated with extracting nectar require less visual acuity. Instead, colour recognition may be a more important component of the visual system for nectarivores when identifying flowers [59,60]. Vertebrate capture was strongly associated with large eyes, likely because capture requires long-distance detection as employed

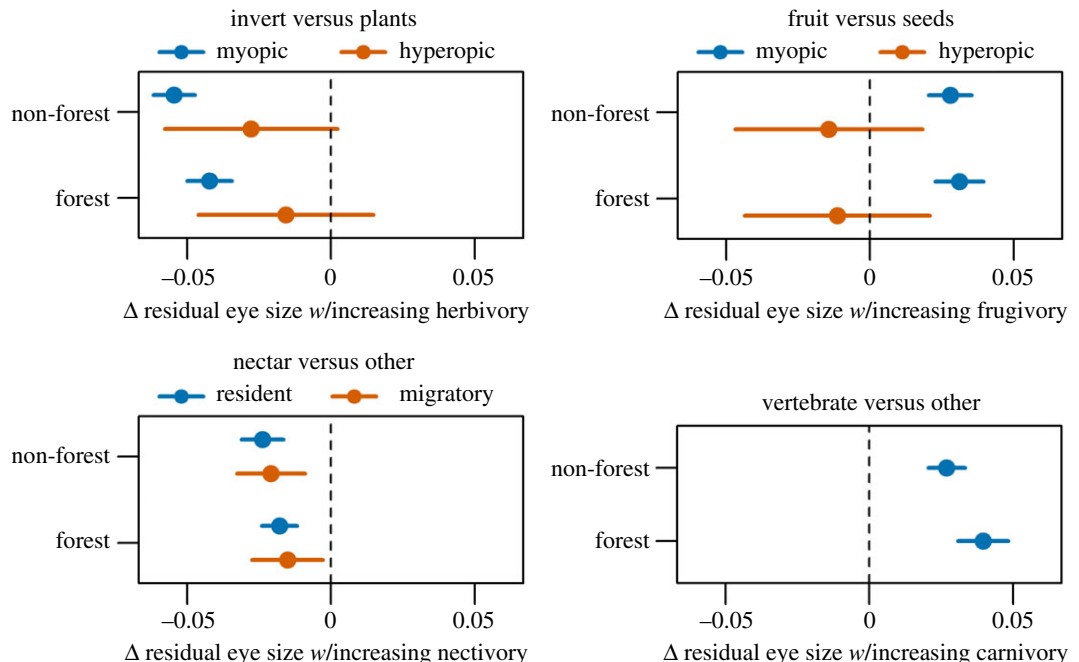

**Figure 3.** Estimated marginal means (+/− 95% CI) of model coefficients for diet in relation to residual eye size for 2777 species of terrestrial birds.

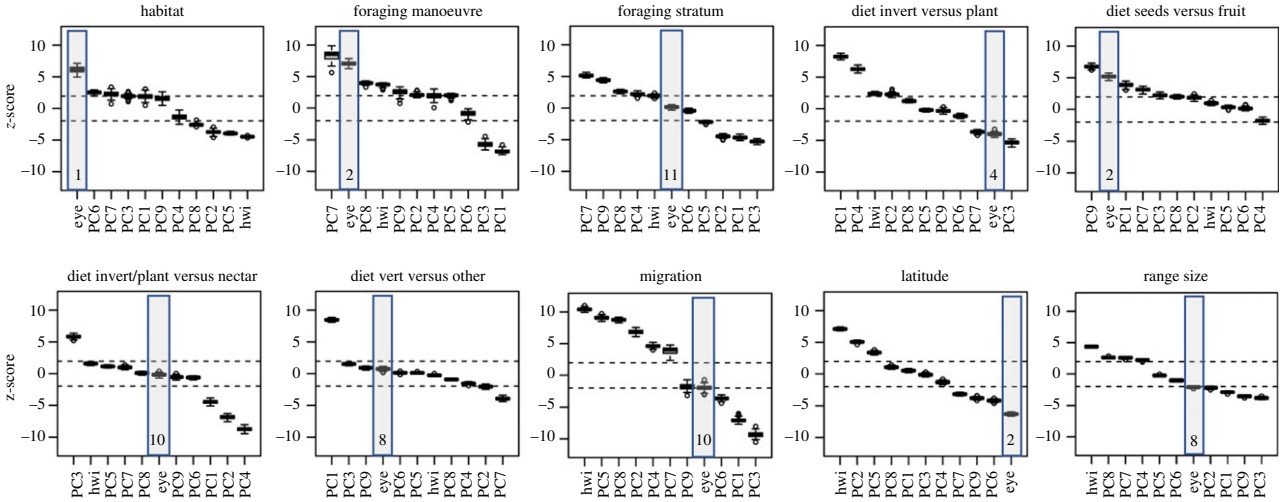

**Figure 4.** z-scores of residual eye size and 10 previously published morphological axes as predictors for 10 ecological variables from phylogenetic regression for 2777 species of terrestrial birds. Boxplots represent variation in z-scores across 100 trees. Scores are significant ($p > 0.05$) outside the dashed lines. Eye size is highlighted with its predictive rank relative to the other morphological axes.

by raptorial species [14]. In sum, small eyes appeared related mainly to nectivory and granivory and large eyes with insectivorous, frugivorous and carnivorous diets.

## (c) Habitat and macro-ecology

The dramatic variation in optical environments found across a species' range requires visual adaptations that optimize survival [61,62]. Species adapted to forest interiors have been hypothesized to have larger eyes (i.e. 'dim forest hypothesis') [32], yet studies remain inconclusive on the associations between eye size and habitat [33,39,63]. This study confirms that forest specialists have larger eyes, supporting the idea that species adapted to extremely dark environments may respond negatively to abrupt spectral changes. Interestingly, eye size was inversely correlated to range size for migratory and non-forest taxa. Species employing these life-history strategies likely encounter the widest variation in light conditions across the terrestrial avifauna, suggesting that smaller eyes are more adaptive

to increased habitat niche breadth [64,65]. Hence, the relative lack of light may constrain minimum eye size for species occupying the dimmest conditions, whereas abrupt transitions to bright light may pose a constraint in maximum eye size for species occupying heterogeneous habitats.

## (d) Ecomorphological comparisons

Avian ecomorphological relationships have long been established for traits related to movement (wing length and shape, tail and tarsus), diet (bill length, depth and width) and size (mass) [16,66]. Given the links between morphological form and function, morphology is often used to explain rates of species diversification and trait evolution [67–69], population-level genetic differentiation [70], niche expansion and packing [8] and functional collapse in disturbed ecosystems [71]. Here, I demonstrate that eye size provides strong relative predictive power for explaining variation in traits related to habitat, trophic niche and life history beyond

Proc. R. Soc. B 288: 20210853

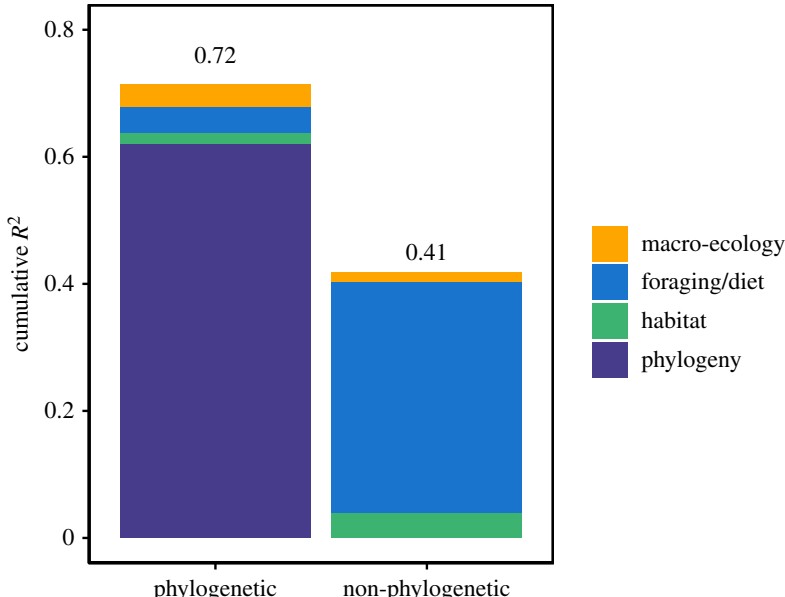

**Figure 5.** Phylogenetic distribution of residual eye size (TD) for a third of terrestrial avian diversity ($N = 2777$ species). Select families are noted. Photo attributions: A. Morffew, A. Radhac, B. Matsubara, B. McCauley, D. Coetzee, D. Church, D. Greenberg, D. Sutherland, F. Veronesi, F. Franklin, G. Smith, J. West, J. Thompson, J. Harrison, J. Boone, Kuribo, L. Boyle, L. Docker, M. Thompson, Mike's Birds, N. Borrow, P. Gaines, P. Kavanagh, T. Benson, T. Wilberding, S. Price, USFWS. All photos are held under CC-BY licenses.

**Figure 6.** Variance partitioning for phylogenetic and non-phylogenetic models of residual eye size for 2777 species of terrestrial birds.

previously published ecomorphological trait axes. In particular, eye size was the strongest single predictive trait in determining whether a species specialized in forest and was a top predictive trait for foraging manoeuvre, diet and latitude. Despite recent exhaustive research describing the ecomorphological relationships for the complete avian tree of life, 22% of variation in the avian foraging niche remains unexplained [16], and eye morphology may contribute towards resolving such residual variation.

## (e) Role of phylogeny

Phylogeny played a predominate role in explaining eye size variation, and residual eye size was highly conserved throughout the avian phylogeny. Despite many significant relationships between ecological variables and eye size, ecological factors explained far less overall variation after controlling for phylogeny. This mirrors evidence that trait evolution has a predictive accuracy of 65% in explaining ecomorphological relationships across the avian tree of life, with contemporary adaptations explaining a further 20% [16]. Results from this study are strikingly similar, attributing a comparable proportion of variance in eye size to ancestral relationships (61%) and recent adaptation (9%). Collectively, high phylogenetic signal in both residual eye size and correlated dietary and foraging traits suggests that light has contributed to the evolution of avian lineages, producing families with markedly different eye morphologies and providing tantalizing evidence of correlated evolution between eye morphology and traits related to the foraging niche.

## (f) Conservation implications

Morphology is often interpreted as a set of functional traits that contribute to the disassembly of avian diversity in anthropogenic landscapes [9,72]. Strong predictive power and correlations with forest specialization and latitude suggest that the relatively dark light micro-environments found within forests, especially in the tropics, contribute towards species-specific sensitivity to habitat disturbance [73]. Given that altered light intensity regimes have been linked to avian community disassembly [33,39], light may act as an environmental filter in anthropogenic landscapes, and morphological adaptations to light should be considered when assessing interspecific sensitivity to habitat fragmentation and land use conversion.

Data accessibility. All datasets and scripts are available from the Dryad Digital Repository: https://doi.org/10.5061/dryad.1c59zw3v9 [74].

Author contributions. IA: conceptualization, data curation, formal analysis, funding acquisition, investigation, methodology, project administration, software, validation, visualization, writing-original draft and writing-review and editing.

Competing interests. I declare I have no competing interests.

Funding. The Katherine Ordway Chair in Ecosystem Conservation at the Florida Museum of Natural History provided funding for data entry.

Acknowledgements. I thank S. Montgomery and K. Perez for digitizing the eye data. R. Kimball, M. Miyamoto, F. Newell, S. Robinson and two anonymous referees provided comments that improved the manuscript. I am indebted to S. Ritland for the heroic amount of work that went into obtaining the eye measurements.

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
