## [Peer Review File · Proceedings of the Royal Society B: Biological Sciences]

Review History

RSPB-2020-2491.R0 (Original submission)

Review form: Reviewer 1

Recommendation

Major revision is needed (please make suggestions in comments)

Scientific importance: Is the manuscript an original and important contribution to its field?

Excellent

General interest: Is the paper of sufficient general interest?

Good

Quality of the paper: Is the overall quality of the paper suitable?

Acceptable

Is the length of the paper justified?

Yes

Should the paper be seen by a specialist statistical reviewer?

No

Do you have any concerns about statistical analyses in this paper? If so, please specify them explicitly in your report.

Yes

It is a condition of publication that authors make their supporting data, code and materials available - either as supplementary material or hosted in an external repository. Please rate, if applicable, the supporting data on the following criteria.

Is it accessible?

Yes

Is it clear?

Yes

Is it adequate?

Yes

Do you have any ethical concerns with this paper?

No

Comments to the Author

This manuscript uses an unpublished dataset of eye measurements from 2,804 avian species from 143 families to examine the ecological correlates of unusually large eyes within a phylogenetic context. The author finds that size-corrected eye width correlates with habitat type and foraging mode, as well as diet, migration, and latitude.

This paper is interesting and generally well-written; it is a novel question asked at an impressive scale, testing a comprehensive set of hypotheses. I especially like the integration with questions of conservation and of large-scale form-function evolution. I have a number of important criticisms, but I think they can all be very easily addressed, and once they are fixed I believe this will be a very strong contribution to the macroevolutionary and macroecological literature.

My main sticking point is the inclusion of non-phylogenetic analyses. I am struggling to understand why they were included; lines 323-325 do not help. It is – and has been for decades – widely established that comparative correlations without phylogenetic correction are simply incorrect; as such, including them adds little of scientific value (and might confuse readers unfamiliar with phylogenetic comparative approaches). The only justification I myself can come up with is a historical one, if it has been very common to run non-phylogenetic analyses on this type of data and the author wishes to emphasize just how misleading that approach is. If this is the case, though – or if there is some other reason for including the non-phylogenetic analyses that I just failed to understand – this needs to be made much clearer in-text.

In a similar vein, I do not understand, at all, what is being done in lines 183-186. What biological question does this analysis address? Can the author perhaps cite a source where this approach has been elsewhere employed?

And for my final major concern – but with recognition that addressing this in the analysis may be well beyond the scope of the paper – does the author think there might be differences in eye size between the crepuscular and diurnal birds in the sample?

Minor comments:

L21-23: The second half of this sentence does not follow from the first half. Strong phylogenetic conservatism could imply several things (e.g. little link to ecological or environmental drivers), but it's hard to jump from any of those interpretations to a "critical role" in avian evolution.

L109-113: Which of the foraging strategies in Pigot et al. 2020 (ref. 15) did you map to myopic and which did you map to hyperopic? How, precisely, was Birds of the World used to supplement this classification system? More details are needed.

L134-136: I found the use of double logs to be surprising, considering how standard $\log(\text{mass})$ is in comparative studies. Are the key results different if only $\log(\text{TD})$ and $\log(\text{mass})$ are used? Could the author cite other works that use this double logarithmic transformation in a similar allometric context?

L162: $\log(\text{mass})$ or $\log(\log(\text{mass}))$? (And thank you for including this paragraph in general, as this was exactly what I was wondering at this point!)

L190: For ease of reader navigation, I would suggest, if the results remain labelled as “Part 1/2/3”, that the methods also be labelled in a parallel fashion as Part 1/2/3. (Or that the exact language is used in the subtitles between the two sections.)

L205 (and elsewhere): “v.” is for legal documents – in all other contexts, the abbreviation is “vs.”

L270-271: My understanding, though, was that raptors were excluded from your study?

L300-302: While I found this main point of this paragraph generally interesting, I’m not sure I agree that a linear relationship between morphological principal component axes and eye size would be expected. (As opposed to, say, body mass or HWI, where the physical interpretation might be clearer.) I would recommend that the author either expand upon why such a correlation might occur, or to temper the language interpreting this lack of simple correlation.

Figure 2: The interpretations of “a”, “ab”, etc., are non-obvious and should be explained in the legend.

Figure 4: The placement of the phylo/non-phylo key in the top-left panel confused me – I think it would be better placed not in a plotting space, or at least with some sort of box around it.

Table S4: How were these interaction terms selected? From the text it sounds like the interactions tested were all binary variables interacted with each of the continuous variables, but from this table that seems not to be the case? And why were some not included in the non-phylogenetic regressions?

Figure S6: Am I reading this correcting that Pagel’s lambda was calculated for some categorical variables (e.g. foraging manoeuvre, migratory tendency)? Pagel’s lambda is generally only for continuous traits. Careful Googling suggests there are extensions for discrete traits (e.g. <https://rdrr.io/github/stoufferlab/phyloint/man/phylo.signal.html> or <https://www.mail-archive.com/r-sig-phylo@r-project.org/msg00922.html>), or for binary traits you can use Fritz and Purvis D.

Review form: Reviewer 2

Recommendation

Accept with minor revision (please list in comments)

Scientific importance: Is the manuscript an original and important contribution to its field?

Acceptable

General interest: Is the paper of sufficient general interest?

Acceptable

Quality of the paper: Is the overall quality of the paper suitable?

Good

Is the length of the paper justified?

Yes

Should the paper be seen by a specialist statistical reviewer?

No

Do you have any concerns about statistical analyses in this paper? If so, please specify them explicitly in your report.

No

It is a condition of publication that authors make their supporting data, code and materials available - either as supplementary material or hosted in an external repository. Please rate, if applicable, the supporting data on the following criteria.

Is it accessible?

Yes

Is it clear?

Yes

Is it adequate?

Yes

Do you have any ethical concerns with this paper?

No

Comments to the Author

General Comments:

This manuscript uses Stanley Ritland's breathtakingly large dataset (2804 bird species) to determine the relationship between eye size and several ecological traits + phylogeny. The methods are well carried out and the conclusions are, by-and-large, sound. I remain slightly concerned about using the eye's transverse diameter a proxy for the eye's axial length (which itself is a proxy for focal length), although removing raptorial species from the sample set went a long way toward assuaging that concern. My primary issues all occurred in the Introduction section, but all together this is a high quality manuscript.

Line-by-line Comments:

Line 54-56: "Hence, eye size likely reflects an evolutionary trade-off between maximizing visual acuity while minimizing overexposure". I disagree with this statement. Eye size has neurological and energetic trade-offs, and trade-offs with other structures for space in the skull (e.g., the brain), that are probably more important than its trade-off with exposure. And there are more pragmatic workarounds for overexposure without sacrificing eye size (e.g., optical adnexa and pupil characteristics) than there are for the other trade-offs that limit eye size.

Line 56-57: I do agree with this prediction. However, this prediction could feasibly (and even likely) remain true without the hypothesis about overexposure. Which is an issue with the hypothesis-prediction relationship, in itself.

Line 62-63: "Because the visual field tends to increase with eye size". To be blunt, this is not a true statement. Geometrically, there is nothing about a small or large eye that inherently limits visual field width. If anything, the observed relationship would be the opposite of the stated direction (but that is due to indirect factors). All of the subsequent examples can be explained by the

positive relationship between eye size and visual acuity, rather than visual field.

Line 64-66: There is a new paper that briefly discusses this topic (Senzaki et al. 2020, Nature)

Line 79: The predictions in lines 80-82 are good. But line 79 is just a more general prediction rather than a hypothesis.

Line 84-86: This is a very interesting prediction. What is the author's accompanying hypothesis?

Methods: The methods are well executed. Very robust sampling decisions.

Line 190-195: These two sentences are redundant.

Line 199: "foraging" rather than "forging"

Line 272: It is interesting that nectivory and frugivory did not fall the same way... could the explanation of small eyes for nectivory not also apply to frugivory?

Decision letter (RSPB-2020-2491.R0)

22-Dec-2020

Dear Mr Ausprey:

I am writing to inform you that your manuscript RSPB-2020-2491 entitled "Adaptations to Light Contribute to the Ecological Niches and Evolution of the Terrestrial Avifauna" has, in its current form, been rejected for publication in Proceedings B.

This action has been taken on the advice of referees, who have recommended that substantial revisions are necessary. With this in mind we would be happy to consider a resubmission, provided the comments of the referees are fully addressed. However please note that this is not a provisional acceptance.

Sincerely,

Dr Daniel Costa

Reviewer(s)' Comments to Author:

Referee: 1

Comments to the Author(s)

This manuscript uses an unpublished dataset of eye measurements from 2,804 avian species from 143 families to examine the ecological correlates of unusually large eyes within a phylogenetic context. The author finds that size-corrected eye width correlates with habitat type and foraging mode, as well as diet, migration, and latitude.

This paper is interesting and generally well-written; it is a novel question asked at an impressive scale, testing a comprehensive set of hypotheses. I especially like the integration with questions of conservation and of large-scale form-function evolution. I have a number of important criticisms, but I think they can all be very easily addressed, and once they are fixed I believe this will be a very strong contribution to the macroevolutionary and macroecological literature.

My main sticking point is the inclusion of non-phylogenetic analyses. I am struggling to understand why they were included; lines 323-325 do not help. It is – and has been for decades – widely established that comparative correlations without phylogenetic correction are simply incorrect; as such, including them adds little of scientific value (and might confuse readers unfamiliar with phylogenetic comparative approaches). The only justification I myself can come up with is a historical one, if it has been very common to run non-phylogenetic analyses on this type of data and the author wishes to emphasize just how misleading that approach is. If this is the case, though – or if there is some other reason for including the non-phylogenetic analyses that I just failed to understand – this needs to be made much clearer in-text.

In a similar vein, I do not understand, at all, what is being done in lines 183-186. What biological question does this analysis address? Can the author perhaps cite a source where this approach has been elsewhere employed?

And for my final major concern – but with recognition that addressing this in the analysis may be well beyond the scope of the paper – does the author think there might be differences in eye size between the crepuscular and diurnal birds in the sample?

Minor comments:

L21-23: The second half of this sentence does not follow from the first half. Strong phylogenetic conservatism could imply several things (e.g. little link to ecological or environmental drivers), but it's hard to jump from any of those interpretations to a "critical role" in avian evolution.

L109-113: Which of the foraging strategies in Pigot et al. 2020 (ref. 15) did you map to myopic and which did you map to hyperopic? How, precisely, was Birds of the World used to supplement this classification system? More details are needed.

L134-136: I found the use of double logs to be surprising, considering how standard $\log(\text{mass})$ is in comparative studies. Are the key results different if only $\log(\text{TD})$ and $\log(\text{mass})$ are used? Could the author cite other works that use this double logarithmic transformation in a similar allometric context?

L162: $\log(\text{mass})$ or $\log(\log(\text{mass}))$? (And thank you for including this paragraph in general, as this was exactly what I was wondering at this point!)

L190: For ease of reader navigation, I would suggest, if the results remain labelled as "Part 1/2/3", that the methods also be labelled in a parallel fashion as Part 1/2/3. (Or that the exact language is used in the subtitles between the two sections.)

L205 (and elsewhere): “v.” is for legal documents – in all other contexts, the abbreviation is “vs.”

L270-271: My understanding, though, was that raptors were excluded from your study?

L300-302: While I found this main point of this paragraph generally interesting, I’m not sure I agree that a linear relationship between morphological principal component axes and eye size would be expected. (As opposed to, say, body mass or HWI, where the physical interpretation might be clearer.) I would recommend that the author either expand upon why such a correlation might occur, or to temper the language interpreting this lack of simple correlation.

Figure 2: The interpretations of “a”, “ab”, etc., are non-obvious and should be explained in the legend.

Figure 4: The placement of the phylo/non-phylo key in the top-left panel confused me – I think it would be better placed not in a plotting space, or at least with some sort of box around it.

Table S4: How were these interaction terms selected? From the text it sounds like the interactions tested were all binary variables interacted with each of the continuous variables, but from this table that seems not to be the case? And why were some not included in the non-phylogenetic regressions?

Figure S6: Am I reading this correcting that Pagel’s lambda was calculated for some categorical variables (e.g. foraging manoeuvre, migratory tendency)? Pagel’s lambda is generally only for continuous traits. Careful Googling suggests there are extensions for discrete traits (e.g. <https://rdrr.io/github/stoufferlab/phyloint/man/phylo.signal.html> or <https://www.mail-archive.com/r-sig-phylo@r-project.org/msg00922.html>), or for binary traits you can use Fritz and Purvis D.

Referee: 2

Comments to the Author(s)

General Comments:

This manuscript uses Stanley Ritland’s breathtakingly large dataset (2804 bird species) to determine the relationship between eye size and several ecological traits + phylogeny. The methods are well carried out and the conclusions are, by-and-large, sound. I remain slightly concerned about using the eye’s transverse diameter a proxy for the eye’s axial length (which itself is a proxy for focal length), although removing raptorial species from the sample set went a long way toward assuaging that concern. My primary issues all occurred in the Introduction section, but all together this is a high quality manuscript.

Line-by-line Comments:

Line 54-56: “Hence, eye size likely reflects an evolutionary trade-off between maximizing visual acuity while minimizing overexposure”. I disagree with this statement. Eye size has neurological and energetic trade-offs, and trade-offs with other structures for space in the skull (e.g., the brain), that are probably more important than its trade-off with exposure. And there are more pragmatic workarounds for overexposure without sacrificing eye size (e.g., optical adnexa and pupil characteristics) than there are for the other trade-offs that limit eye size.

Line 56-57: I do agree with this prediction. However, this prediction could feasibly (and even likely) remain true without the hypothesis about overexposure. Which is an issue with the hypothesis-prediction relationship, in itself.

Line 62-63: “Because the visual field tends to increase with eye size”. To be blunt, this is not a true statement. Geometrically, there is nothing about a small or large eye that inherently limits visual field width. If anything, the observed relationship would be the opposite of the stated direction (but that is due to indirect factors). All of the subsequent examples can be explained by the positive relationship between eye size and visual acuity, rather than visual field.

Line 64-66: There is a new paper that briefly discusses this topic (Senzaki et al. 2020, Nature)

Line 79: The predictions in lines 80-82 are good. But line 79 is just a more general prediction rather than a hypothesis.

Line 84-86: This is a very interesting prediction. What is the author's accompanying hypothesis?

Methods: The methods are well executed. Very robust sampling decisions.

Line 190-195: These two sentences are redundant.

Line 199: "foraging" rather than "forging"

Line 272: It is interesting that nectivory and frugivory did not fall the same way... could the explanation of small eyes for nectivory not also apply to frugivory?

Author's Response to Decision Letter for (RSPB-2020-2491.R0)

See Appendix A.

RSPB-2021-0853.R0

Review form: Reviewer 1

Recommendation

Accept with minor revision (please list in comments)

Scientific importance: Is the manuscript an original and important contribution to its field?

Excellent

General interest: Is the paper of sufficient general interest?

Good

Quality of the paper: Is the overall quality of the paper suitable?

Excellent

Is the length of the paper justified?

Yes

Should the paper be seen by a specialist statistical reviewer?

No

Do you have any concerns about statistical analyses in this paper? If so, please specify them explicitly in your report.

No

It is a condition of publication that authors make their supporting data, code and materials available - either as supplementary material or hosted in an external repository. Please rate, if applicable, the supporting data on the following criteria.

Is it accessible?

Yes

Is it clear?

Yes

Is it adequate?

Yes

Do you have any ethical concerns with this paper?

No

Comments to the Author

This revised manuscript is a beautifully written study which uses an unpublished dataset of eye size measurements from 2,777 species of birds and phylogenetic comparative methods to demonstrate that habitat, foraging strategy, diet, and latitude all correlate with variation in avian eye morphology. I reviewed a previous version of this manuscript, and I believe the current version to be greatly improved by the thoughtful and thorough responses to my and the other reviewer's comments. The methods and results are clearly explained, the dataset is incredible in its depth, and I find the suggested links to community assemblage and conservation to be both interesting and appropriately measured.

I have three microscopic comments; otherwise, I congratulate the author on an excellent piece of work!

L92-94: I am not quite sure what you mean by this paragraph. Consider omitting? Or rephrasing to make it clearer why one might predict high explanatory power for phylogeny in a dataset such as this? That high phylogenetic signal suggests high explanatory power for phylogenetic correction is not really a prediction; it's just the definition of phylogenetic signal. Perhaps I am missing something?

L201: Missing close quote on 'fitDiscrete'.

Ref 73: Something's gone slightly amiss with the citation here.

Decision letter (RSPB-2021-0853.R0)

19-Apr-2021

Dear Mr Ausprey

I am pleased to inform you that your manuscript RSPB-2021-0853 entitled "Adaptations to Light Contribute to the Ecological Niches and Evolution of the Terrestrial Avifauna" has been accepted for publication in Proceedings B.

The referee(s) have recommended publication, but also suggest some minor revisions to your manuscript. Therefore, I invite you to respond to the referee(s)' comments and revise your manuscript. Because the schedule for publication is very tight, it is a condition of publication that you submit the revised version of your manuscript within 7 days. If you do not think you will be able to meet this date please let us know.

[http://datadryad.org/submit?journalID=RSPB&manu=\(Document not available\)](http://datadryad.org/submit?journalID=RSPB&manu=(Document not available)) which will take you to your unique entry in the Dryad repository. If you have already submitted your data to dryad you can make any necessary revisions to your dataset by following the above link.

Please see <https://royalsociety.org/journals/ethics-policies/data-sharing-mining/> for more details.

Sincerely,
Dr Daniel Costa
mailto: proceedingsb@royalsociety.org

Reviewer(s)' Comments to Author:

Referee: 1

Comments to the Author(s).

This revised manuscript is a beautifully written study which uses an unpublished dataset of eye size measurements from 2,777 species of birds and phylogenetic comparative methods to demonstrate that habitat, foraging strategy, diet, and latitude all correlate with variation in avian eye morphology. I reviewed a previous version of this manuscript, and I believe the current version to be greatly improved by the thoughtful and thorough responses to my and the other reviewer's comments. The methods and results are clearly explained, the dataset is incredible in its depth, and I find the suggested links to community assemblage and conservation to be both interesting and appropriately measured.

I have three microscopic comments; otherwise, I congratulate the author on an excellent piece of work!

L92-94: I am not quite sure what you mean by this paragraph. Consider omitting? Or rephrasing to make it clearer why one might predict high explanatory power for phylogeny in a dataset such as this? That high phylogenetic signal suggests high explanatory power for phylogenetic correction is not really a prediction; it's just the definition of phylogenetic signal. Perhaps I am missing something?

L201: Missing close quote on 'fitDiscrete'.

Ref 73: Something's gone slightly amiss with the citation here.

Author's Response to Decision Letter for (RSPB-2021-0853.R0)

See Appendix B.

Decision letter (RSPB-2021-0853.R1)

20-Apr-2021

Dear Mr Ausprey

I am pleased to inform you that your manuscript entitled "Adaptations to Light Contribute to the Ecological Niches and Evolution of the Terrestrial Avifauna" has been accepted for publication in Proceedings B.

Data Accessibility section

Open Access

Paper charges

Sincerely,

Appendix A

Referee: 1

Comments to the Author(s)

This manuscript uses an unpublished dataset of eye measurements from 2,804 avian species from 143 families to examine the ecological correlates of unusually large eyes within a phylogenetic context. The author finds that size-corrected eye width correlates with habitat type and foraging mode, as well as diet, migration, and latitude.

This paper is interesting and generally well-written; it is a novel question asked at an impressive scale, testing a comprehensive set of hypotheses. I especially like the integration with questions of conservation and of large-scale form-function evolution. I have a number of important criticisms, but I think they can all be very easily addressed, and once they are fixed I believe this will be a very strong contribution to the macroevolutionary and macroecological literature.

Thank you for the kind words and encouragement.

My main sticking point is the inclusion of non-phylogenetic analyses. I am struggling to understand why they were included; lines 323-325 do not help. It is – and has been for decades – widely established that comparative correlations without phylogenetic correction are simply incorrect; as such, including them adds little of scientific value (and might confuse readers unfamiliar with phylogenetic comparative approaches). The only justification I myself can come up with is a historical one, if it has been very common to run non-phylogenetic analyses on this type of data and the author wishes to emphasize just how misleading that approach is. If this is the case, though – or if there is some other reason for including the non-phylogenetic analyses that I just failed to understand – this needs to be made much clearer in-text.

While ancestral relationships violate statistical assumptions of independence among species, those relationships in themselves are of biological interest. For this reason, I compared the variation in eye size explained both with and without phylogenetic relationships to explicitly quantify the degree to which eye size reflects evolutionary processes apart from ecology. Except for the section on explained variation, I have taken your advice and removed the non-phylogenetic results from all parts of the manuscript as they added little value and varied minorly compared to the phylogenetic results. I have retained the results for Figure 6 (variation explained by phylogeny), because it dramatically demonstrates that the vast majority of variation in eye size is due to phylogeny. This publication touches on why this type of analysis is interesting:

<https://academic.oup.com/sysbio/article/68/2/234/5098616>

I have removed the text from lines 323-325 in the original draft.

In a similar vein, I do not understand, at all, what is being done in lines 183-186. What biological question does this analysis address? Can the author perhaps cite a source where this approach has been elsewhere employed?

I have altered this analysis to focus only on phylogenetic signal of the 10 ecological variables and clarified my explanation.

And for my final major concern – but with recognition that addressing this in the analysis may be well beyond the scope of the paper – does the author think there might be differences in eye size between the crepuscular and diurnal birds in the sample?

Thank you for this point. I have removed 27 species and four families with crepuscular foraging behaviors. This changed the total sample size of species to N = 2777.

Minor comments:

L21-23: The second half of this sentence does not follow from the first half. Strong phylogenetic conservatism could imply several things (e.g. little link to ecological or environmental drivers), but it's hard to jump from any of those interpretations to a "critical role" in avian evolution.

I have removed this strong language.

L109-113: Which of the foraging strategies in Pigot et al. 2020 (ref. 15) did you map to myopic and which did you map to hyperopic? How, precisely, was Birds of the World used to supplement this classification system? More details are needed.

I have added language in the Methods and added a new table to the supplementary material that maps out the classification from Pigot et al. 2020.

L134-136: I found the use of double logs to be surprising, considering how standard $\log(\text{mass})$ is in comparative studies. Are the key results different if only $\log(\text{TD})$ and $\log(\text{mass})$ are used? Could the author cite other works that use this double logarithmic transformation in a similar allometric context?

I have reanalyzed the data for this revision based on $\log(\text{TD}) \sim \log(\text{mass})$. This did not change results.

L162: $\log(\text{mass})$ or $\log(\log(\text{mass}))$? (And thank you for including this paragraph in general, as this was exactly what I was wondering at this point!)

This has been changed.

L190: For ease of reader navigation, I would suggest, if the results remain labelled as "Part 1/2/3", that the methods also be labelled in a parallel fashion as Part 1/2/3. (Or that the exact language is used in the subtitles between the two sections.)

Done.

L205 (and elsewhere): "v." is for legal documents – in all other contexts, the abbreviation is "vs."

Done.

L270-271: My understanding, though, was that raptors were excluded from your study?

Yes, raptors were excluded. However, many non-raptorial species incorporate substantial amounts of vertebrate prey in their diets (e.g., shrikes, rollers, etc.). The dietary PC scores for these species will

reflect the contribution of carnivorous aspects of their diet.

L300-302: While I found this main point of this paragraph generally interesting, I'm not sure I agree that a linear relationship between morphological principal component axes and eye size would be expected. (As opposed to, say, body mass or HWI, where the physical interpretation might be clearer.) I would recommend that the author either expand upon why such a correlation might occur, or to temper the language interpreting this lack of simple correlation.

I have removed "is correlated" from this sentence and rewritten it to better emphasize the more important point, that eye size provides additional explanatory power. I have also included a new figure of the correlation matrix between eye size and the 9 trait axes in the supplement to show the shape of the correlations.

Figure 2: The interpretations of "a", "ab", etc., are non-obvious and should be explained in the legend.

Done.

Figure 4: The placement of the phylo/non-phylo key in the top-left panel confused me – I think it would be better placed not in a plotting space, or at least with some sort of box around it.

This has been re-formatted.

Table S4: How were these interaction terms selected? From the text it sounds like the interactions tested were all binary variables interacted with each of the continuous variables, but from this table that seems not to be the case? And why were some not included in the non-phylogenetic regressions?

Yes, interactions were coded between just binary and continuous variables for the initial "global" model. Table S4 just shows the variables and interactions retained from the stepwise AIC procedure. Not all binary x continuous variable interactions were retained by the stepwise procedure and used in the final "ecological" model. I have included new language in the Methods to clarify this.

Figure S6: Am I reading this correcting that Pagel's lambda was calculated for some categorical variables (e.g. foraging manoeuvre, migratory tendency)? Pagel's lambda is generally only for continuous traits. Careful Googling suggests there are extensions for discrete traits (e.g.

<https://rdrr.io/github/stoufferlab/phyloint/man/phylo.signal.html> or <https://www.mail-archive.com/r-sig-phylo@r-project.org/msg00922.html>), or for binary traits you can use Fritz and Purvis D.

Thank you for the helpful suggestion. I have reanalyzed signal of the three binary traits using the 'fitDiscrete' function from the package 'geiger'. The approach you cite above is based upon this function.

Referee: 2

Comments to the Author(s)

General Comments:

This manuscript uses Stanley Ritland's breathtakingly large dataset (2804 bird species) to determine the

relationship between eye size and several ecological traits + phylogeny. The methods are well carried out and the conclusions are, by-and-large, sound. I remain slightly concerned about using the eye's transverse diameter a proxy for the eye's axial length (which itself is a proxy for focal length), although removing raptorial species from the sample set went a long way toward assuaging that concern. My primary issues all occurred in the Introduction section, but all together this is a high quality manuscript.

Thank you for helpful and encouraging feedback. I initially limited the analysis to transverse diameter (TD) as Stanley Ritland had written that he was less confident in his axial diameter (AD) measurements – indeed, his original Dissertation focuses almost entirely on TD. However, based on your comments, I have analyzed the data for AD as well. Both TD and AD provide similar results. Results for AD are provided in the supplement.

Line-by-line Comments:

Line 54-56: "Hence, eye size likely reflects an evolutionary trade-off between maximizing visual acuity while minimizing overexposure". I disagree with this statement. Eye size has neurological and energetic trade-offs, and trade-offs with other structures for space in the skull (e.g., the brain), that are probably more important than its trade-off with exposure. And there are more pragmatic workarounds for overexposure without sacrificing eye size (e.g., optical adnexa and pupil characteristics) than there are for the other trade-offs that limit eye size.

I have added extensive language discussing tradeoffs associated with metabolic investment, the brain, and flight aerodynamics to the Introduction. I have retained the language on overexposure, because there is anatomical and experimental evidence for "disability glare":

<https://www.karger.com/Article/Abstract/47218>

<https://www.sciencedirect.com/science/article/pii/S0003347207001960?via%3Dihub>

Line 56-57: I do agree with this prediction. However, this prediction could feasibly (and even likely) remain true without the hypothesis about overexposure. Which is an issue with the hypothesis-prediction relationship, in itself.

I have updated this sentence to include multiple factors that could contribute to the evolutionary tradeoff.

Line 62-63: "Because the visual field tends to increase with eye size". To be blunt, this is not a true statement. Geometrically, there is nothing about a small or large eye that inherently limits visual field width. If anything, the observed relationship would be the opposite of the stated direction (but that is due to indirect factors). All of the subsequent examples can be explained by the positive relationship between eye size and visual acuity, rather than visual field.

I apologize – this was incorrectly written. It has been changed to "visual acuity".

Line 64-66: There is a new paper that briefly discusses this topic (Senzaki et al. 2020, Nature)

I have added a sentence on artificial light and avian reproduction.

Line 79: The predictions in lines 80-82 are good. But line 79 is just a more general prediction rather than a hypothesis.

I have completely rewritten the hypotheses and predictions to make them more specific.

Line 84-86: This is a very interesting prediction. What is the author's accompanying hypothesis?

I place the prediction regarding tropical forests under the new habitat hypothesis to more explicitly link the idea to my hypothesis about habitat and light.

Methods: The methods are well executed. Very robust sampling decisions.

Thank you (credit goes to Stanley Ritland).

Line 190-195: This two sentences are redundant.

I moved the second sentence to the start of the Discussion.

Line 199: "foraging" rather than "forging"

Done.

Line 272: It is interesting that nectivory and frugivory did not fall the same way... could the explanation of small eyes for nectivory not also apply to frugivory?

This result is due to the nature of the dietary principal component axes. The second dietary principal component axis is positively correlated with increased frugivory and negatively correlated with granivory; hence, eye size increases with frugivory ONLY compared to granivory. The third dietary principal component axis is positively correlated with nectivory and moderately negatively correlated with invertebrate, fruit, and seed dietary components; hence, eye size decreases with nectivory compared to frugivory (at least partially).

Appendix B

Referee: 1

Comments to the Author(s).

This revised manuscript is a beautifully written study which uses an unpublished dataset of eye size measurements from 2,777 species of birds and phylogenetic comparative methods to demonstrate that habitat, foraging strategy, diet, and latitude all correlate with variation in avian eye morphology. I reviewed a previous version of this manuscript, and I believe the current version to be greatly improved by the thoughtful and thorough responses to my and the other reviewer's comments. The methods and results are clearly explained, the dataset is incredible in its depth, and I find the suggested links to community assemblage and conservation to be both interesting and appropriately measured.

I have three microscopic comments; otherwise, I congratulate the author on an excellent piece of work!

Thank you for the kind praise.

L92-94: I am not quite sure what you mean by this paragraph. Consider omitting? Or rephrasing to make it clearer why one might predict high explanatory power for phylogeny in a dataset such as this? That high phylogenetic signal suggests high explanatory power for phylogenetic correction is not really a prediction; it's just the definition of phylogenetic signal. Perhaps I am missing something?

This paragraph is now deleted.

L201: Missing close quote on 'fitDiscrete'.

Corrected.

Ref 73: Something's gone slightly amiss with the citation here.

Corrected.